# A New Function for Amyloid-Like Interactions: Cross-Beta Aggregates of Adhesins form Cell-to-Cell Bonds

**DOI:** 10.3390/pathogens10081013

**Published:** 2021-08-11

**Authors:** Peter N. Lipke, Marion Mathelié-Guinlet, Albertus Viljoen, Yves F. Dufrêne

**Affiliations:** 1Biology Department, Brooklyn College of City University of New York, 2900 Bedford Avenue, Brooklyn, NY 11210, USA; 2Louvain Institute of Biomolecular Science and Technology, Université catholique de Louvain, Croix du Sud, 4-5, bte L7.07.07, B-1348 Louvain-la-Neuve, Belgium; marion.mathelie@uclouvain.be (M.M.-G.); albertus.viljoen@uclouvain.be (A.V.); yves.dufrene@uclouvain.be (Y.F.D.)

**Keywords:** *Candida albicans*, *Saccharomyces cerevisiae*, biofilm, adhesin, protein conformation, AFM, steric zipper, nanodomain

## Abstract

Amyloid structures assemble through a repeating type of bonding called “cross-β”, in which identical sequences in many protein molecules form β-sheets that interdigitate through side chain interactions. We review the structural characteristics of such bonds. Single cell force microscopy (SCFM) shows that yeast expressing Als5 adhesin from *Candida albicans* demonstrate the empirical characteristics of cross-β interactions. These properties include affinity for amyloid-binding dyes, birefringence, critical concentration dependence, repeating structure, and inhibition by anti-amyloid agents. We present a model for how cross-β bonds form *in trans* between two adhering cells. These characteristics also apply to other fungal adhesins, so the mechanism appears to be an example of a new type of cell–cell adhesion.

## 1. Introduction

We have recently discovered that amyloid-like bonds form between cells in fungal biofilms [1,2]. These bonds show properties of cross-β aggregation, like the bonds formed between amyloid proteins in neurodegenerative diseases [3,4,5]. These intercellular bonds are in addition to the well-characterized interactions that cluster the adhesion proteins (the adhesins) on yeast cell surfaces [6]. Among the functional consequences are discoveries that some anti-amyloid compounds can prevent biofilm formation. We review the evidence for amyloid-like cell–cell adhesion. This novel type of bond between arrays of adhesion proteins has profound implications for the energetics and mechanical resistance of the cell–cell bonds, and this discovery extends to roles in pathogenesis and biofilm formation. If this new type of cell-to-cell bond formation proves to be general [6,7,8,9], there are important implications for the treatment of invasive fungal infections and for cell adhesion and histogenesis in other fungi and in animals [10,11,12,13].

## 2. Cross-β Aggregation and Amyloids

Amyloids are fibrous aggregates of proteins with a unique type of non-covalent bonding called cross-β aggregation. In these structures, identical amino acids in parallel or anti-parallel arrays form β sheets, and these sheets then adhere through amino acid side chain interactions and interdigitations in a structure called a steric zipper (Figure 1A) [3,4,14]. Although individual β-sheets have poor stability in aqueous environments, stacks of sheets stabilized by sidechain–sidechain interactions are extremely stable. Thus, stacked β-sheets with steric zipper interactions are the structural basis of the amyloid fibrils of many proteins. These cross-β interactions are a major subset of molecular interactions in diseases that are characterized by inappropriate aggregation and phase separations [15]. Most cross-β aggregates are insoluble and are often almost irreversible. Thus, they are notorious as markers of neurodegenerative diseases (including Aβ fibrils in Alzheimer’s disease, alpha-synuclein aggregates in Parkinson’s disease, prions in Creutzfeldt–Jakob Disease and scrapie, and others [4,16,17]). Interestingly, as opposed to these pathogenic amyloids, there are also functional amyloids, in which cross-β aggregates are essential for function of proteins [18,19,20,21,22]. These include β-helical interactions in virus tail spikes, bacterial curlins and *Pseudomonas* biofilm PAF proteins in biofilms of Gram-negative bacteria, and in Gram-positive Staphylococcal PSMs [23,24,25,26]. In mammalian systems, melanosomes are amyloid assemblies, as are condensing granules for some peptide hormone precursors in the pituitary gland [19,27,28]. 

Both pathogenic and functional cross-β aggregates have a set of characteristics commonly used for their identification as amyloid-like structures (Figure 1). These include a dependence on sequence (the sequence in each strand must be identical or highly similar), birefringence between crossed polaroids, binding of amyloidophilic dyes like Congo red and thioflavins, rigid fibers of 8–20 nm diameter visible by transmission electron microscopy, and a characteristic X-ray diffraction pattern in fiber diffraction [3,4]. Amyloid fibers are formed around ‘core sequences’ of five to seven successive amino acid residues that are identifiable through a variety of search algorithms that measure solubility, geometry, and/or sequence similarity [29,30,31,32,33]. 

## 3. Fungal Adhesins, Amyloid Fibrils, and Cross-β Aggregates 

Many fungal adhesins show these properties, and some can form amyloid fibers in vitro (Figure 1 and Figure 2). The *Candida albicans* Als adhesins are the best characterized, but the list also includes the *Saccharomyces cerevisiae* Flo1 and Flo11 adhesins. Peptides with the core sequences from these adhesins and many others can form amyloid fibers in vitro (Figure 1B) [34,35]. The Als and Flo adhesins themselves also form amyloid fibers in vitro [6,34]. However, in vivo, the adhesins are covalently anchored to cell wall glycans, and so they cannot form fibers the way that soluble proteins can. Instead, the adhesins form surface patches or ‘nanodomains’ of ~50–100 nm in diameter, as visualized by atomic force microscopy and confocal microscopy (Figure 2). These patches have the properties of cross-β structures, including sequence dependence, thioflavin binding, birefringence, and inhibition by anti-amyloid treatments. Nanodomain formation increases the avidity of cell-to-cell interactions because the clustering of adhesins drives up the local concentration of the binding sites on adhesins, so any ligands are likely to bind to any of many nearby binding sites [6].

## 4. Cross-β Bonding *in Trans*

In addition, recent AFM experiments with intact live cells reveal that cell-to-cell bonds themselves show cross-β bond characteristics, which is consistent with amyloid-like bonding between cells. In single cell force spectroscopy (SCFS), an individual live cell is attached to an AFM tipless cantilever resulting in a so-called cell probe, which is then lowered until it contacts another live cell trapped in a porous membrane and the AFM stage (Figure 3). As the probe is then retracted away from the sample surface, adhesion forces are recorded if any adhesive bond occurs between the cell probe and the targeted cell. Such SCFS investigations have allowed for quantitative demonstrations of the strong adhesion between Als5p-expressing cells. This cell-to-cell adhesion increases while successively mapping the interactions between cells of the same pair as force activates the increasing formation of adhesin nanodomains through β-aggregation-mediated clustering and the subsequent formation of cell-to-cell bonds through similar cross-β aggregation [1,2].

For *C. albicans* Als5 adhesin, the cell–cell interactions show amyloid-like properties (i.e., the interaction depends on the presence of the amyloid core sequence ^325^IVIVATT^331^ on both interacting cells (Figure 3A vs. Figure 3B) [2]). Like amyloid formation, cell–cell adhesions are dependent on high local concentrations of the amyloid core sequences, and therefore there must be high concentrations of the adhesins on the cell surfaces. Acute concentration dependence is a defining characteristic for cross-β structures because multiple identical sequences form the β-strands that comprise the β-sheets, and multiple β-sheets need to interact through steric zippers. Thus, the concentration dependence is due to the need for enough molecules to constitute multiple β-sheets, an assembly necessary to stabilize the cross-β structure [3,4,14]. A supporting study shows that amyloid-like β-interactions form *in trans* between an amyloidogenic peptide attached to the AFM tip and an Als5p molecule on the AFM stage surface. In these experiments, the same amyloid sequence must be also present in both the peptide and the Als5 adhesin [6]. Additionally, cell–cell binding is inhibited by treatments that disrupt cross-β amyloid-like bonds. Thus, these data, along with previous findings that thioflavin-bright patches form at the sites of cell–cell adhesion (Figure 2A), support a model in which amyloid-like bonds form *in trans* between the cells (Figure 4).

## 5. Biogenesis of Cross-β Bonds *in Trans*

A model for the biogenesis of these structures is illustrated in Figure 4. Initially, shear force unfolds the partially structured T domains in Als proteins, exposing the amyloid core sequences (1). These sequences then mediate assembly of adhesins into nanodomain arrays on the surface of a cell *in cis* (2), nanodomains that can exist with either parallel-strand β-sheets (right), or mixed parallel and anti-parallel strands (left). Finally, the nanodomains interact *in trans* through homologous cross-β aggregation between two cells (3, orange–brown, and blue). Steps 2 and 3 may generate different orientations of the arrays. Either orientation or a mixed set of orientations would be consistent with the data so far.

Figure 4 shows only a single β-sheet layer; a model of two interacting sheets in a *trans* array is shown in Figure 5A. In vivo the *trans* arrays are based on hundreds to thousands of H bonds and steric zipper interactions between the cells, and so should be highly stable. The zipper requires homologous interactions to maintain both the intra- and inter-sheet associations, so sequence identity is key. Thus, these bonds are expected to be strong and highly stable, as observed in assays in vitro [6,37,38]. These characteristics were recognized and named ‘SRS adhesions’ for strong reversible specific two decades ago [37]. We now know the molecular basis for these characteristics.

Formation of cross-β structures is slow, because of the need for extensive conformational changes and the entropic costs of alignment [39]. Furthermore, interacting proteins need to remain in contact with each other during bond formation to ‘lock-in’ the cross-β structures as they form [39]. Clearly, cell-to-cell bond formation is facilitated by prior assembly of the *cis* nanodomains, which are already in the appropriate conformation. Both nanodomain formation and cell–cell adhesion have halftimes of about 7 min, and this observation implies that nanodomain formation is a rate-limiting step [8,40,41]. Nanodomain assembly is probably facilitated by prior non-amyloid interactions of the adhesins. These interactions include hydrophobic effect clustering through the Tandem Repeats [42] and Ig-domain binding to peptide epitopes in other Als molecules. Ig-domain ligands include the three-residue sequence motif called τϕ+ (tau, a residue common in turns; phi, a bulky hydrophobic residue; and +: Lys or Arg) [42]. Thus, initial hydrophobic effect leads to binding of τϕ+ motifs between adhesins, followed by formation of the cross-β aggregates. Only this last step is inhibited by anti-amyloid treatments.

## 6. Other Adhesins

How general is this model of β-aggregation *in trans* for cell–cell adhesion? *C. albicans* Als1 shares the same sequences of the amyloid core and T domains with Als5, rendering their similarities highly predictable. Experimental studies have indeed confirmed that both adhesins show the same characteristics of cell-to-cell binding initiated by shear-mediated [6,38,43,44,45]. Further, cell-to-cell adhesion occurs with the highest probability and strength when both cells express Als1 with the wild-type cross-β core sequence, while cells that express the non-amyloid substitution V326N have reduced adhesion probability and strength [38]. This points to a β-aggregation *in trans* as described above. As in Als5, adhesin clustering and cell–cell adhesion are inhibited in the presence of thioflavin S or the anti-amyloid peptide SNGINIVATRTV. Thus, Als1-mediated homophilic adhesion follows the model for Als5 in its reliance on cross-β bond formation.

Data on the *S. cerevisiae* flocculins Flo1 and Flo11 also support the formation of cross-β bonds *in trans* between expressing cells. Like the Als proteins, these adhesins also have multiple binding modes. The flocculins are partially Ca^2+^-dependent, but they can be activated by shear force in the absence of Ca^2+^ [42,46,47]. This result is consistent with the idea that cation binding induces a conformational change that helps to expose amyloid-like core sequences.

The Flo1 family has Ca^2+^-dependent lectin (glycan binding) activity mediated by the N-terminal PA14 domain [46,47,48]. The glycan-binding N-terminal domain is followed by a variable number of ~50 residue tandem repeats. These repeats show no sequence similarity to the Als repeats, but as in Als proteins, the repeats are predicted to have anti-parallel β-sheet structure, and the Flo1 repeats unfold under extension or shear force [6,49]. Unlike the Als adhesins, each Flo1 repeat contains an amyloid core sequence with high frequency of hydrophobic aliphatic residues. Thus, unfolding leads to exposure of many homologous amyloid-core sequences. As expected for β-aggregation *in trans*, alleles with more repeats have stronger binding [50,51]. Flocculation can be prevented or reversed by millimolar concentrations of the amyloid dye thioflavin S [52]. Furthermore, Flo1-mediated aggregates are birefringent between crossed polaroids, as are Als5 aggregates. A second activity of Flo1 (its ability to invade agar) is also inhibited by thioflavin S or by Congo red. In addition, Flo1-expressing colonies recognize other Flo1-expressing cells as ‘self’ in forming biofilms and exclude cells that do not express Flo1 [53]. This ‘greenbeard’ property (the ability to recognize individuals who will help to form a biofilm, and to exclude ‘cheaters’ who will not adhere strongly and thus will not help form the biofilm) is consistent with the idea of cross-β-dependent cell–cell bonding. It is unlikely to be based on the lectin activity of the protein because all yeast cells are coated with the lectin ligand α-mannosides [53,54]. Thus, the properties of Flo1-medated aggregation are like Als5 and are consistent with the idea of cross-β bonds formed *in trans* between cells.

Flo11 is a *S. cerevisiae* adhesin with no sequence similarity to Flo1, but it also shows properties consistent with cross-β bonding *in trans*. The Flo11 N-terminal domain (NTD) can self-associate through homologous interactions of belts of Trp residues [55,56,57]. There are potential amyloid core sequences in the post-NTD and C-terminal regions [34]. Like Flo1 and Als adhesins, the ability to form cellular aggregates is induced by shear force. Flocculation depends on both cells expressing Flo11, and so depends on homophilic interactions [57,58,59,60]. The flocs are birefringent [6]. Thioflavin S and Congo red inhibit flocculation and Flo11-mediated agar invasion. As in Flo1, these properties are consistent with cross-β interactions between cells being critical for activity of the adhesin.

In our hands, these cell surface cross-β aggregates are not reversible, probably because aggregate disassembly and domain refolding are both thermodynamically unfavored. Thus, the remodeling of these cell associations would probably require that trans-aggregates be shed from cell surfaces by proteolysis. In support of this speculation, dibasic residues are common in known and potential adhesins with cross-β core sequences [7]. Proteolytic shedding would destroy the trans-cellular adhesions and would also generate extracellular amyloid-like structures made of the shed material. That outcome may be desirable in biofilms where amyloids can become part of the matrix, as in bacterial biofilms [7,24,25,61,62,63,64]. In support of this idea, other wall proteins also include cross-β core sequences [32,65,66,67]. Among these, the glycosyl transferases Gas1, Gas3, Gas 5, Ygp1, and Bgl2 can form intracellular β-aggregates, and form amyloids in *E. coli* when overexpressed in the Csg-curlin system [22,66,67,68,69]. There are also other putative adhesins that show cross-β core sequences embedded in regions with low sequence complexity with little stable structure. For these candidates, we have not yet evidence for cross-β bonding in situ or for a biological activity or consequence.

Similarly, β-aggregation core sequences are present in mammalian adhesion molecules, including integrins, CAMs and EpCAM, cadherins, and lectins, but currently none are known to be functional [70]. There are intriguing hints about the mammalian synaptogenic cell adhesion molecule APLP-1 [71]. Like the Flo adhesins, it clusters in a metal dependent manner at sites of cell–cell contact. Fluorescence transfer experiments show close association of APLP-1 molecules *in trans* across cell contacts. Two β-aggregation core sequences are present in E2 coiled-coil domain of APLP-1, and as in the adhesins, this domain would need to unfold to allow assembly of a β-aggregate. However, there is no evidence (yet) that the β-aggregation core sequences are involved in APLP-1-mediated cell–cell binding *in trans.*

## 7. Summary

β-aggregation *in trans* as a cell–cell adhesion mode is newly discovered. In *C. albicans* Als adhesins and *S. cerevisiae* flocculation, cross-β aggregates form after core sequences are exposed by shear-stress-induced unfolding of protein domains. The result is formation of inter-cellular aggregates and shear-resistant biofilms. Structural data probably awaits further developments in cryo-EM technology that would allow structural determination of molecular complexes in the intercellular space. Thus, the consequences of cross-β assembly *in trans* are just now beginning to be clear, and discovering these consequences will lead to new ways to think about cell adhesion and about biofilms.

## Figures and Tables

**Figure 1 pathogens-10-01013-f001:**
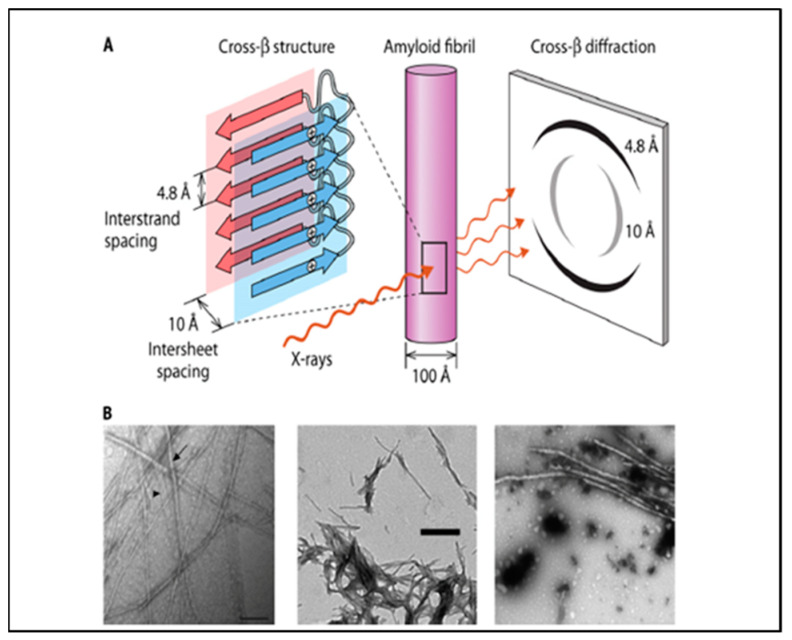
Amyloid structures. (**A**) Arrangement of β-strands in an amyloid fiber (a cross-β structure). The figure shows two β-sheets with one of the identical residues in each strand marked with a ‘+’ in a circle. The residues are aligned in each strand. This cartoon is shown as part of an amyloid fiber in the middle, and the characteristic “cross-β” fiber X-ray diffraction pattern is on the right. (**B**) Amyloid fibers from fungal adhesins: left to right are peptides *Candida albicans* adhesin Eap1, a tridecameric peptide from Als5, and a 644-residue fragment of Als5. Reprinted from [6].

**Figure 2 pathogens-10-01013-f002:**
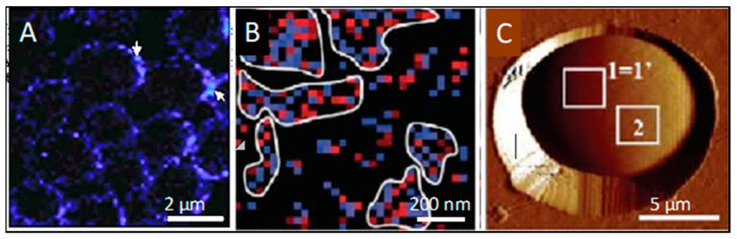
Cross-β surface nanodomains on live yeast cells. (**A**) Confocal image of thioflavinT-labelled nanodomains on yeast cells. Arrows point to bright spots at cell–cell adhesions. (**B**) A map of V5-labelled Als5 in a 1µm square on the surface of a living yeast cell. V5-epitope-labelled adhesin Als5 was expressed on the surface of *Saccharomyces cerevisiae,* then probed with an AFM tip derivatized with anti-V5. Pixels are colored where the AFM detects a V5 epitope: blue for rupture forces <150 pN and red for rupture forces ≥150 pN. After mechanical stimulation, the adhesins are clustered. (**C**) Yeast cell embedded in a micro-porous membrane on an AFM stage. The cell is expressing a V5-tagged version of Als5 on its surface, and the map of the area marked “1 = 1’” is image B. Reprinted from [36].

**Figure 3 pathogens-10-01013-f003:**
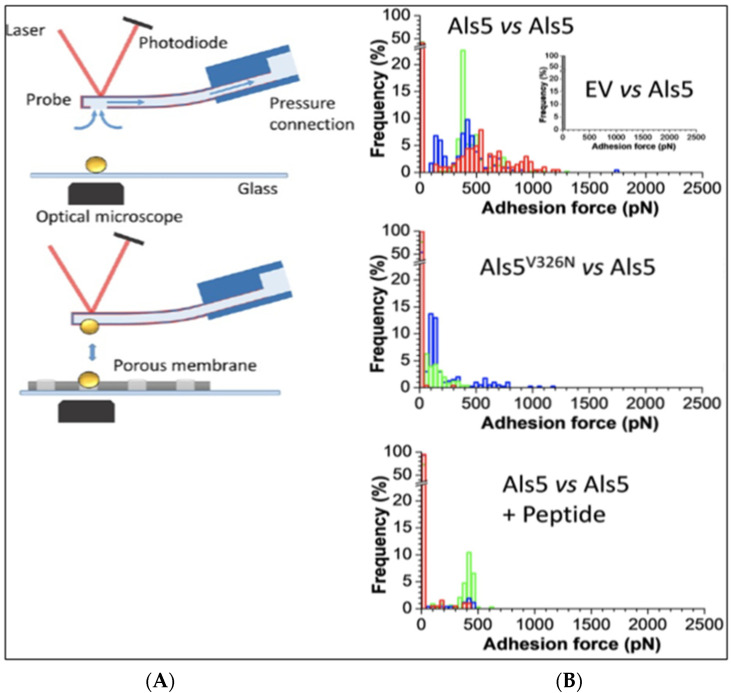
Single cell force microscopy (SCFM). (**A**) A cell is attached to the AFM tip and brought into contact with another cell on the AFM stage. (**B**) Force histograms for maximum cohesive force between cell pairs upon retraction of the cell attached to the AFM tip. Frequency and strength of adhesion are maximal when both cells express the cross-β-forming version Als5 (top histogram). The center panel shows lower frequency of adhesion and force when one cell expresses the non-amyloid V326N variant. The bottom panel shows a force histogram of pairs of cells expressing the cross-β forming version of the adhesin in the presence of a sequence-specific anti-amyloid peptide. Reprinted from [2].

**Figure 4 pathogens-10-01013-f004:**
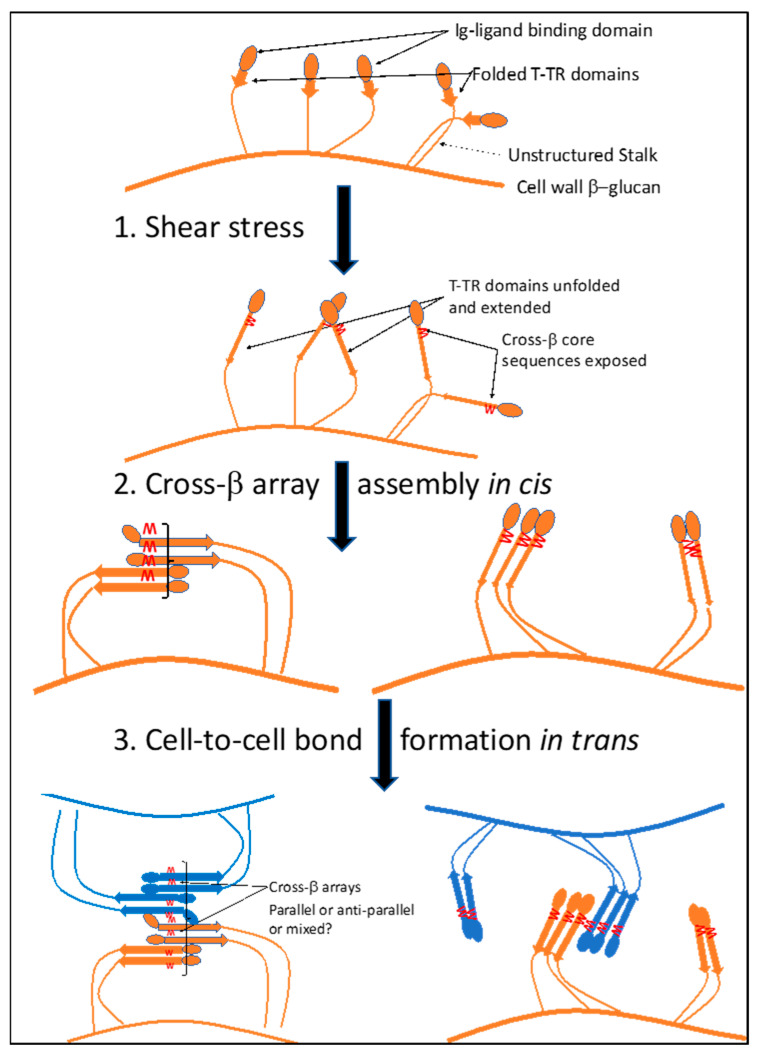
Model for formation of cross-β bonded Als5 *cis*-nanodomains and *trans*-adhesions on the surface of a cell. In step 1, shear force unfolds the T domain, exposing the cross-β core sequence in each molecule (red W symbols). In step 2, the exposed core sequences assemble into *cis*-nanodomains as constrained by the length of the flexible stalk region of the proteins. Two possible orientations are shown. In step 3, the *cis*-nanodomains associate *in trans* to form cell-to-cell bonds. Again, two possible orientations are shown; either orientation is compatible with the data so far.

**Figure 5 pathogens-10-01013-f005:**
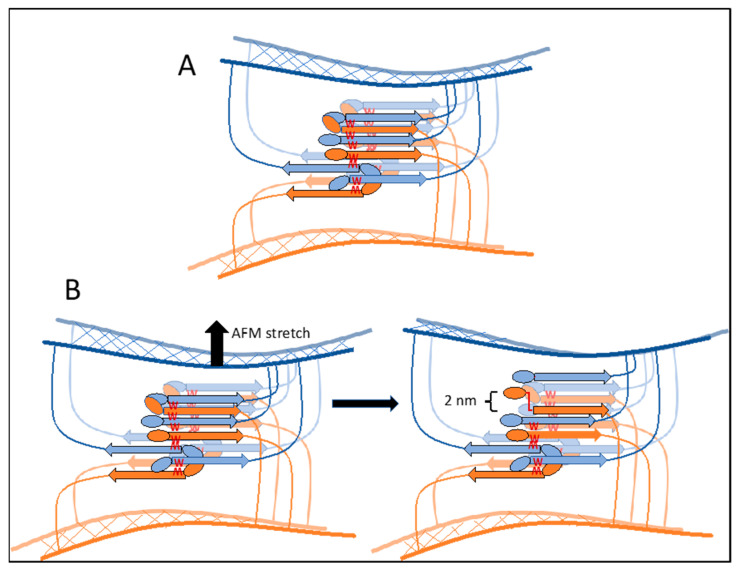
Two-sheet model cross β cell–cell bonding *in trans*. (**A**) A model showing the interaction of β-sheets in the cross-β bonds between two cells, one cell in orange–brown and the other in blue. A β-sheet in front is outlined and brighter colored than the sheet behind. (**B**) A model showing an origin for the 2 nm extension quantum observed when cells are separated. In SCFM, as cells are separated (top arrow), the most stressed bond breaks, and the cross-β core sequence is stretched to its limit (~2 nm for a 6 amino acid sequence). The cells remain associated through other bonds. Subsequently, the next-most stressed bond will dissociate, and the process repeats to generate the successive 2 nm separation peaks observed in the force–distance curves [2].

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
