# Peer review of "A New Function for Amyloid-Like Interactions: Cross-Beta Aggregates of Adhesins form Cell-to-Cell Bonds"

_pathogens, 2021, doi:10.3390/pathogens10081013_

Round 1

Reviewer 1 Report

This is a well-written review article about an interesting and important topic.  The author provides an excellent background review on the topic of cross-beta aggregate formation and its role in fungal cell-to-cell adhesion.   The only suggestion I would make is to provide a little more information in the figure legends to help the reader better understand the information being conveyed in the figures.  

Author Response

We thank the reviewer for the suggestion, and have expanded all the figure legends.

Reviewer 2 Report

The manuscript at hand reviews what is known about a new role of amyloid-like interactions connected to intracellular clustering of adhesins and intercellular aggregation. As such, it is an interesting subject which deserves being reviewed and the authors are competent and experts in the field.

I don’t have severe criticisms, but would suggest a number of editorial improvements as follows:

  • Species names and other words of latin origin (in vivo, in vitro, in situ) should all be written in italics.
  • Please also check reference list for these spellings. In addition, there are some informations missing (e.g. in the first reference there are neither volume nor page numbers (or an E-number, if this was strictly online). Also, there are abbreviations like [pii] and [doi] at the end of most references, which should be omitted.
  • Please state in the legend of Fig. 2 what yeast that is. Is it S. cerevisiae? – As the authors also talk a lot about Candida, I think it is important to give the species name frequently. Also, what is the difference of blue and red pixels in Fig. 2B?
  • Organization of Fig. 4 is not very pleasant. I would suggest to completely omit the letters A, B, C, since they coincide with steps 1-3 discussed in the text. Please don’t forget to rephrase the one sentence where the letters are referred to in the text.
  • Chapter 7: This is a short review and if one reads it carefully, everything listed in chapter 7 is highly redundant and should have been kept in mind from the previous text. Also, this chapter is written in a slightly odd way and at least to me is missing a logical order in the listing and even sometimes missing a word. Even the header seems unusual. I would strongly suggest to omit this chapter entirely, as it does neither contribute new information, nor add to the readability of the review.
  • If chapter 7 is omitted, chapter 8 could be headed “conclusions”, which I think fits perfectly with what is said in there.

Last of all, of the 80 references listed, 23 (!) are self-citations. Though I perfectly understand that the authors contributed significantly to the field, as stated above, this is a bit too much, giving it a touch of autobiography. I would suggest to reduce self-citations to a necessary minimum, say by one-third.

Author Response

We thank the reviewers, and have modified the manuscript to meet the suggestions. It is a better paper as a result.

  1. Italics have been added to species names, gene names, and Latin phrases
  2. References have been checked, corrected, and italics added where appropriate.
  3. The legend to Fig 2 has been expanded to answer these points.
  4. The figure has been clarified and reformatted to try to improve clarity. The index letters have been omitted from the figure, legend, and text.
  5. Section 7 has been removed, and section 8 re-titled and renumbered as suggested
  6. We have removed many references to our primary publications, referring to appropriate reviews instead. This has reduced the self-citations somewhat. However, because there are few other labs working on fungal wall amyloid sequences, most of the information and many of the papers are products of our own research groups.